# Impact of peripheral artery disease on short-term outcomes after percutaneous coronary intervention: A report from Japanese nationwide registry

**Takayuki Ishihara**[1]*, **Kyohei Yamaji**[2], **Osamu Iida**[1], **Shun Kohsaka**[3],
**Taku Inohara**[3], **Toshiro Shinke**[4], **Hirohiko Ando**[5], **Tetsuya Amano**[5], **Yasushi Sakata**[6],
**Toshiaki Mano**[1], **Yuji Ikari**[7]

1 Kansai Rosai Hospital Cardiovascular Center, Amagasaki, Japan, 2 Department of Cardiology, Kokura Memorial Hospital, Kitakyushu, Japan, 3 Department of Cardiology, Keio University School of Medicine, Tokyo, Japan, 4 Division of Cardiovascular Medicine, Department of Internal Medicine, Showa University School of Medicine, Tokyo, Japan, 5 Department of Cardiology, Aichi Medical University, Nagakute, Japan, 6 Department of Cardiovascular Medicine, Osaka University Graduate School of Medicine, Suita, Japan, 7 Department of Cardiology, Tokai University Hospital, Isehara, Japan

☯ These authors contributed equally to this work.
‡ These authors also contributed equally to this work
* t.ishihara31@gmail.com

**Data Availability Statement:** Data are available only upon request according to the "Act on the Protection of Personal Information" Law (as of May 2017) and the "Ethical Guidelines for Medical

## Abstract

Atherosclerosis is a systemic process. As the population ages, increasingly more patients who undergo coronary revascularization are complicated with peripheral artery disease (PAD). However, the large body of evidence in this area has not been limited to analysis from trial-based data from younger and relatively uncomplicated patients in Western countries. The impact of PAD on the outcomes can differ by patient characteristics, and integrated analysis of large-scale data is necessary. J-PCI is a universal (all-comer) nationwide registration system in Japan, regulated and audited by professional society that controls national board-certification system. For the present study, we extracted data of 894,014 percutaneous coronary intervention (PCI) cases performed between 2014 and 2017 (mean age 70.2 years [standard deviation 11.0]). In-hospital outcomes of PAD and Non-PAD patients were compared. PAD was defined as a previous history of stenosis of peripheral arteries or abdominal aortic aneurysm. Primary outcome was in-hospital mortality, and multivariable modeling was performed. A total of 66,891 patients (8.1%) had PAD. Crude in-hospital mortality rate was higher in this group (0.99% vs. 0.67% in Non-PAD group). PAD was associated with an increased risk of in-hospital mortality (odds ratio [OR] 1.383 [95% confidence interval 1.251–1.528]). However, the impact of PAD differed by kidney condition (OR 1.578 [1.370–1.821] for patients with chronic kidney disease [CKD] and OR 1.234 [1.076–1.416] without CKD: P for interaction 0.005), and by clinical presentation: PAD was not associated with an increased risk of in-hospital mortality in patients undergoing PCI for silent ischemia (OR 1.211 [0.8701–1.685]: P for interaction 0.002). Presence of PAD was independently associated with in-hospital mortality in patients receiving PCI. However, its impact varied substantially by the patient background or indication of the procedure.

and Health Research Involving Human Subjects" (as of March 2015). The current study data was obtained from the J-PCI registry and would be available upon request to University of Tokyo, Healthcare Quality Assessment and Japanese Association of Cardiovascular Intervention and Therapeutics Registry Subcommittee (E-mail: info@cvit.jp).

**Funding:** The J-PCI registry is a registry led and supported by the Japanese Association of Cardiovascular Intervention and Therapeutics. KS reports investigator-initiated grant funding from Bayer and Daiichi Sankyo, and personal fees from Bayer, AstraZeneca and Bristol-Myers Squibb. TI has a research grant from Boston Scientific. TS receives Remuneration from Abbott Vascular, Dai-ichi Sankyo, Bayer, Sanofi, and Nipro. TA receives lecture fees from Astellas Pharma, AstraZeneca, Bayer, Daiichi Sankyo, and Bristol-Myers Squibb. YS receives Honorarium from·Otsuka Pharmaceutical, Daiichi Sankyo, Takeda Pharmaceutical,·Mitsubishi Tanabe Pharma, Medtronic Japan and Boehringer Ingelheim Japan, research grant from Edwards Lifesciences, FUJIFILM RI Pharma,·REGiMMUNE, and·Roche Diagnostics, and Scholarship (educational) grant/ endowed chair from Otsuka Pharmaceutical, Johnson & Johnson, St. Jude Medical Japan, Daiichi Sankyo,·Takeda Pharmaceutical,·Mitsubishi Tanabe Pharma, Teijin Pharma Limited,·Boehringer Ingelheim Japan, Bayer Yakuhin, BIOTRONIK Japan, Boston Scientific, Medtronic Japan. TM has a research grant from Abbott vascular Japan. YI has research grants from Boston Scientific, Asahi intech, Nipro, Sanofi, Daiichi-Sankyo, and Terumo, and receives lecture fee from Asteras Amgen Biopharma, Astrazeneca, Abbott vascular, Sanofi, Daiichi-Sankyo, Boehringer-Ingerheim, Bayer, and Bristol Meyers-Squib. The remaining authors have no disclosures to report. The funders had no role in study design, data collection and analysis, decision to publish, or preparation of the manuscript.

**Competing interests:** The authors have read the journal's policies and declare the following competing interests: KS received personal fees from Bayer, AstraZeneca and Bristol-Myers Squibb. TS received funding from Abbott Vascular, Dai-ichi Sankyo, Bayer, Sanofi, and Nipro. TA receives lecture fees from Astellas Pharma, AstraZeneca, Bayer, Daiichi Sankyo, and Bristol-Myers Squibb. YS receives Honorarium from·Otsuka Pharmaceutical, Daiichi Sankyo, Takeda Pharmaceutical,·Mitsubishi Tanabe Pharma, Medtronic Japan and Boehringer Ingelheim Japan, research grant from Edwards Lifesciences, FUJIFILM RI Pharma,·REGiMMUNE,

## Introduction

Atherosclerosis is increasingly becoming recognized as a systemic disorder and peripheral arterial disease (PAD), defined as a diverse group of disorders that lead to progressive stenosis or occlusion, or aneurysmal dilation, of the aorta and its branch arteries, including the carotid, upper extremity, visceral, and lower extremity arterial branches, exclusive of the coronary arteries, is frequently encountered during coronary revascularization in modern practice, particularly in Asian countries where population is aging rapidly [1–4]. From previous studies, patients with coronary artery disease (CAD) are known to have higher risk of cardiovascular events including death if they are complicated with PAD [5,6].

The large body of evidence assessing the impact of PAD on clinical outcomes following percutaneous coronary intervention (PCI) has not been limited to analysis from trial-based data from Western countries with younger and relatively uncomplicated patients. Furthermore, as it has not been elucidated to date whether the impact of PAD varies by the patient background or indication of the procedure, integrated analysis of large-scale data is necessary to solve this issue. Hence, the purposes of the current study, which included one of the largest cohorts of PAD patients who underwent PCI for both acute and elective indications, were as follows; 1) to compare the in-hospital outcomes among PAD and Non-PAD patients, and 2) to evaluate the impact of PAD on in-hospital mortality in clinical-relevant subgroup within a contemporary Japanese nationwide coronary intervention registry.

## Materials and methods

### National clinical data (J-PCI registry) and study design

The J-PCI Registry, which was launched in November 2011, is a national, prospective, multi-center registry designed to collect clinical data on patients undergoing PCI in Japan and is operated by the Japanese Association of Cardiovascular Intervention and Therapeutics. Since January 2013, the J-PCI Registry has been incorporated into the National Clinical Data system, which is a nationwide prospective web-based registry linked to the interventional board certification system [7–9]. The National Clinical Data continuously communicates with data managers responsible for data collection through the National Clinical Data web-based data management system and consistently performs random site visits to validate the submitted data (20 sites/year) [10]. According to the annual reports of the Japanese Registry Of All cardiac and vascular Diseases (JROAD): Annual Report 2016, 1046875 PCIs (284089 PCIs for acute indications and 762786 PCIs for non-acute indications) were performed from January 2014 to December 2017 (http://www.j-circ.or.jp/jittai_chosa/, accessed on March 31, 2019). Given that we included a total of 894014 PCIs over 4 years, around 85% of all procedures in Japan were estimated to be registered in our registry.

The study protocol of the J-PCI registry was approved by the Institutional Review Board Committee at the Network for Promotion of Clinical Studies (a specified nonprofit organization affiliated with Osaka University Graduate School of Medicine [Osaka, Japan]) and complied with the principles contained within the Declaration of Helsinki. Written informed consent was routinely obtained from all patients before undergoing PCI.

### Variables

PAD was defined as a previous history of non-cardiac artery disease, defined as any of the following; stenosis ≥50% of peripheral arteries such as renal, iliac, femoral arteries as well as abdominal aortic aneurysm (AAA) including a symptom of intermittent claudication, ankle brachial index (ABI) ≤0.9 and previous histories of amputation, bypass surgery and

and·Roche Diagnostics, and Scholarship (educational) grant/endowed chair from Otsuka Pharmaceutical, Johnson & Johnson, St. Jude Medical Japan, Daiichi Sankyo,·Takeda Pharmaceutical,·Mitsubishi Tanabe Pharma, Teijin Pharma Limited,·Boehringer Ingelheim Japan, Bayer Yakuhin, BIOTRONIK Japan, Boston Scientific, Medtronic Japan. TM has a research grant from Abbott vascular Japan. YI has research grants from Boston Scientific, Asahi intech, Nipro, Sanofi, Daiichi-Sankyo, and Terumo, and receives lecture fee from Asteras Amgen Biopharma, Astrazeneca, Abbott vascular, Sanofi, Daiichi-Sankyo, Boehringer-Ingerheim, Bayer, and Bristol Meyers-Squib. This does not alter our adherence to PLOS ONE policies on sharing data and materials.

endovascular therapy for the stenosis or occlusion of artery. J-PCI did not require sub-categorization, such as intermittent claudication or critical limb ischemia and details of PAD was not available.

Other recorded clinical characteristics included age, sex, hypertension, diabetes mellitus, current smoking, chronic kidney disease (CKD), hemodialysis, chronic obstructive pulmonary disease, history of PCI, history of coronary artery bypass surgery, history of myocardial infarction, chronic heart failure, clinical presentation at PCI, stent thrombosis within 1 month, general condition within 24 hours, elective PCI, number of disease vessels, access site and radiation time. Hypertension was defined as systolic blood pressure >140 mm Hg, diastolic blood pressure >90 mm Hg, or medical treatment for hypertension. Diabetes mellitus was defined as a fasting plasma glucose level ≥126 mg/dL, a casual plasma glucose level ≥200 mg/dL, a 2-hour plasma glucose level during the 75 g oral glucose tolerance test ≥200 mg/dL, a hemoglobin A1c level >6.5%, or treatment for diabetes mellitus. Dyslipidemia was defined as a total cholesterol level ≥220 mg/dL, a low-density lipoprotein cholesterol level ≥140 mg/dL, a high-density lipoprotein cholesterol level <40 mg/dL, a triglyceride level ≥150 mg/dL, or treatment for hyperlipidemia. CKD was defined as the presence of proteinuria, a serum creatinine level ≥1.3 mg/dL, or an estimated glomerular filtration rate level ≤60 mL/min per 1.73 m$^2$. Stent thrombosis was defined according to the Academic Research Consortium (ARC) definition [11]. Only ARC-definite stent thrombosis was counted in this registry. The definitions of these J-PCI variables are available online (http://www.cvit.jp/registry/jpci_definition.pdf).

## Outcome measures

In the present study, we compared in-hospital outcomes between PAD and Non-PAD patients as a primary outcome. Primary outcome measure was in-hospital mortality, and secondary outcome measures were procedure-related myocardial infarction, cardiac tamponade, cardiogenic shock, stent thrombosis, emergent surgery, bleeding with need for transfusion (related to access site or non-related to access site). Procedure related myocardial infarction was defined as elevation of cardiac troponin values (>5 x 99th percentile URL) according to the Third Universal Definition of Myocardial Infarction [12].

## Statistical analysis

The J-PCI Registry does not include blood pressure or the other numerical variables except for age. All results are expressed as the mean ± standard deviation unless otherwise stated. Continuous variables were analyzed by Welch's t-test. Categorical variables were analyzed with $\chi^2$ test for 2 x 2 comparisons. Multivariable analysis for in-hospital mortality was performed by logistic regression analysis and the following explanatory variables were included: PAD, age, sex, hypertension, diabetes mellitus, dyslipidemia, current smoker, CKD, hemodialysis, previous history of PCI, previous history of coronary artery bypass grafting (CABG), chronic heart failure, previous myocardial infarction, chronic obstructive pulmonary disease, clinical presentation at the time of PCI, cardiac arrest within 24 hours, cardiogenic shock within 24 hours, acute heart failure within 24 hours, access site, and number of diseased vessels. To further assess the impact of PAD on in-hospital mortality, we additionally explored the subgroup analysis in the following strata: age ≥75 years, sex, diabetes mellitus, CKD, hemodialysis, chronic heart failure, clinical presentation at PCI, access site, number of diseased vessels. A two-sided p-value of <0.05 was considered significant in the interaction analysis. All statistical analyses were performed using the use of R version 3.6.1 (R Foundation for Statistical Computing, Vienna, Austria) and python version 3.7.0 (available at https://www.python.org/).

## Results

### Patients

PAD group comprised 66,891 patients (8.1%: Fig 1). In patient characteristics, the rates of comorbidity except for dyslipidemia and current smoker were constantly higher in PAD patients than in Non-PAD patients (Table 1).

In lesion and procedural characteristics, PCI success rate and the usage of drug-eluting stent were lower and the usages of drug-coated balloon and Rotablator were higher in PAD group than in Non-PAD group (Table 2).

### In-hospital outcomes

In-hospital mortality was significantly higher in PAD group than in Non-PAD group (Table 3: 0.99% versus 0.67%). The incidences of procedure-related myocardial infarction, cardiac tamponade, cardiogenic shock, emergent surgery, and bleeding need for transfusion were also significantly higher in PAD group. However, only stent thrombosis was significantly lower in PAD group than in Non-PAD group, although the absolute incidence was very low (0.10% versus 0.15%).

After the adjustment for confounders, PAD was associated with an increased risk of in-hospital mortality (Table 4: odds ratio [OR] 1.383 [95% confidence interval 1.251–1.528]). Similar to PAD, chronic kidney disease (OR 1.493 [1.395–1.598]), cardiac arrest within 24 hours (OR 3.466 [3.195–3.760]), cardiogenic shock within 24 hours (OR 3.909 [3.579–4.268]), acute heart failure within 24 hours (OR 1.957 [1.812–2.115]), and left main disease (OR 2.345 [2.141–2.568]) demonstrated relatively higher odds ratio.

The magnitude of impact of PAD on in-hospital mortality differed according to the presence or absence of CKD (Fig 2: OR 1.579 [1.370–1.821] and OR 1.234 [1.076–1.416] for the CKD and No-CKD subgroup, respectively: P for interaction = 0.005). In addition, PAD was not associated with an increased risk of in-hospital mortality in patients

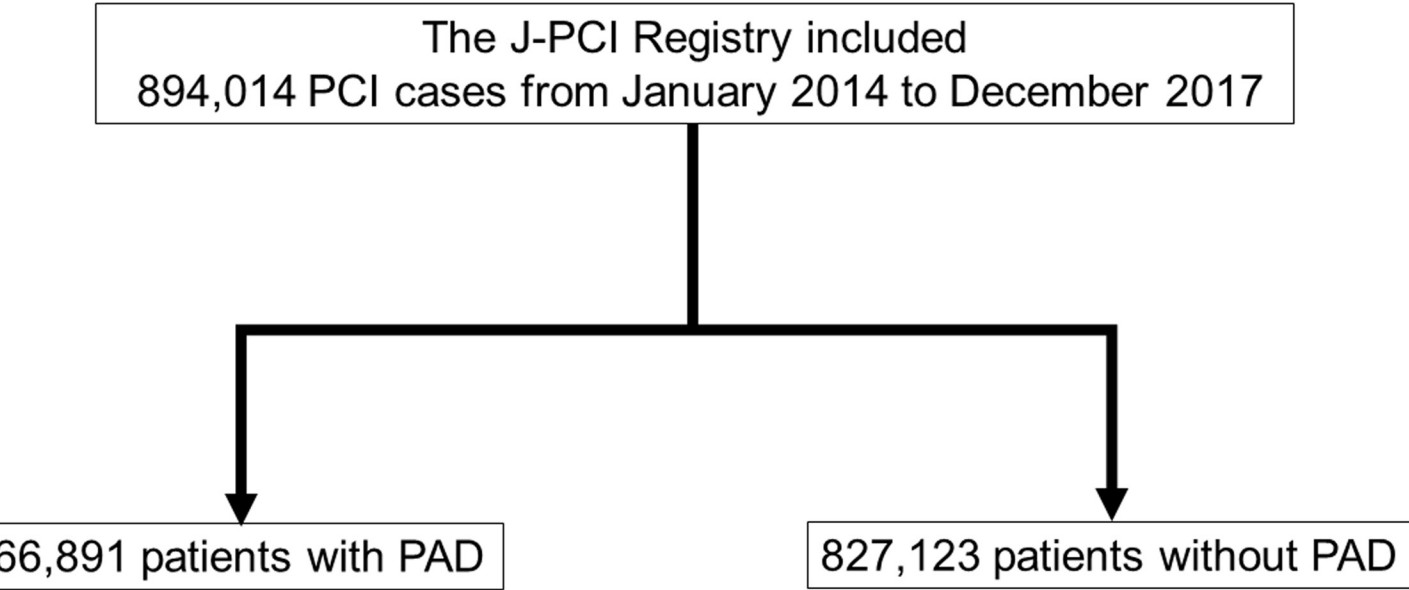

**Fig 1. The study flow chart.** The J-PCI Registry included 894,014 PCI cases from January 2014 to December 2017. Of these, PAD was complicated in 8.1% of the patients. PCI: Percutaneous coronary intervention; PAD: Peripheral artery disease.

**Table 1. Patient characteristics.**

| | PAD group (n = 66,891) | Non-PAD group (n = 827,123) |
|---|---|---|
| Age, y | 73.4±9.06 | 69.9±11.1 |
| Male sex, n (%) | 51,551 (77) | 631,609 (76) |
| Hypertension, n (%) | 54,836 (82) | 640,795 (77) |
| Diabetes mellitus, n (%) | 38,045 (57) | 369,950 (45) |
| Dyslipidemia, n (%) | 42,591 (64) | 549,399 (66) |
| Current smoker, n (%) | 20,917 (31) | 266,551 (32) |
| Chronic kidney disease, n (%) | 25,674 (38) | 132,666 (16) |
| Hemodialysis, n (%) | 14,343 (21) | 46,402 (5.6) |
| Chronic obstructive pulmonary disease, n (%) | 3,798 (5.7) | 15,303 (1.9) |
| Previous history of PCI (n = 883,360), n (%) | 37,149 (56) | 382,529 (47) |
| Previous history of CABG (n = 883,293), n (%) | 5,428 (8.2) | 29,831 (3.7) |
| Old myocardial infarction (n = 878,413), n (%) | 16,082 (24) | 191,005 (24) |
| Chronic heart failure (871,145), n (%) | 15,550 (24) | 104,496 (13) |
| Clinical presentation at PCI (n = 877,618), n (%) | | |
| STEMI | 4,600 (7.0) | 148,199 (18) |
| NSTEMI | 2,170 (3.3) | 38,824 (4.8) |
| Unstable angina pectoris | 8,940 (14) | 127,965 (16) |
| Stable angina pectoris | 25,192 (38) | 308,092 (38) |
| Old myocardia. infarction | 3,612 (5.5) | 47,124 (5.8) |
| Silent ischemia | 20,950 (32) | 141,950 (17) |
| Stent thrombosis within 1 month, n (%) | 179 (0.27) | 2,561 (0.31) |
| General condition within 24 hours, n (%) | | |
| Cardiac arrest (n = 877,254) | 912 (1.4) | 13,062 (1.6) |
| Cardiogenic shock (n = 877,090) | 1,826 (2.7) | 24,089 (3.0) |
| Acute heart failure (n = 876,826) | 2,575 (3.9) | 31,406 (3.9) |
| Elective PCI | 57,685 (86) | 600,275 (73) |
| Number of diseased vessels | | |
| 1 vessel | 34,653 (52) | 502,043 (61) |
| 2 vessels | 17,894 (27) | 201,039 (24) |
| 3 vessels | 10,171 (15) | 91,896 (11) |
| Left main disease | 4,173 (6.2) | 32,145 (3.9) |
| Access site | | |
| Femoral artery | 4,435 (37) | 239,109 (29) |
| Radial artery | 34,370 (51) | 547,648 (66) |
| Others | 8,086 (12) | 40,358 (4.9) |
| Radiation time (n = 771,029), min | 33.2±26.8 | 29.5±24.0 |

CABG: Coronary artery bypass grafting; NSTEMI: Non-ST elevation myocardial infarction; PAD: Peripheral artery disease; PCI: Percutaneous coronary intervention; STEMI: ST elevation myocardial infarction.

undergoing PCI for silent ischemia (OR 1.211 [0.8701–1.685]), whereas it consistently increased the risk of in-hospital mortality in patients treated with PCI for ST elevation myocardial infarction (OR 1.249 [1.072–1.456], non-ST elevation myocardial infarction (OR 1.522 [1.191–1.945]), unstable angina pectoris (OR 1.361 [1.066–1.739]), stable angina pectoris (OR 1.788 [1.292–2.473]) and old myocardial infarction (OR 2.630 [1.596–4.335], P for interaction = 0.002).

**Table 2. Lesion and procedural characteristics.**

| | All (n = 1,348353) | PAD group (n = 108,435) | Non-PAD group (n = 1,239,918) |
|---|---|---|---|
| PCI success (n = 1,347,441), n (%) | 1,316,941 (98) | 105,312 (97) | 1,211,629 (98) |
| Target vessel, n (%) | | | |
| Left main-left anterior descending artery | 584,951 (43) | 40,947 (38) | 544,004 (44) |
| Left circumflex artery | 274,621 (20) | 22,617 (21) | 252,004 (20) |
| Right coronary artery | 433,998 (32) | 39,199 (36) | 394,799 (32) |
| Left main trunk | 48,766 (3.6) | 4,838 (4.5) | 43,928 (3.5) |
| Bypass graft | 4,705 (0.35) | 705 (0.65) | 4,000 (0.32) |
| Others | 1,312 (0.097) | 129 (0.12) | 1,183 (0.095) |
| Drug-coated balloon, n (%) | 100,907 (7.5) | 10,891 (10) | 90,016 (7.3) |
| Bare-metal stent, n (%) | 33,430 (2.5) | 2,015 (1.9) | 31,415 (2.5) |
| Drug-eluting stent, n (%) | 106,0210 (79) | 82,443 (76) | 977,767 (79) |
| Bioabsorbable scaffold, n (%) | 1,073 (0.080) | 100 (0.092) | 973 (0.078) |
| Rotablator, n (%) | 48,233 (3.6) | 7,463 (6.9) | 40,770 (3.3) |
| Directional atherectomy, n (%) | 1,248 (0.093) | 72 (0.066) | 1,176 (0.095) |

PAD: Peripheral artery disease; PCI: Percutaneous coronary intervention.

## Discussion

To the best of our knowledge, this is the first large-scale report outlining the impact of PAD on in-hospital mortality in patients undergoing PCI especially for each subgroup. The current study has demonstrated the following: 1) 8.1% of patients who received PCI were complicated with PAD according to this nationwide PCI registration system in Japan; 2) the crude in-hospital mortality rate was significantly higher in patients with PAD than in those without PAD (0.99% versus 0.57%); 3) PAD was an independent predictor for in-hospital mortality, and notably, had the stronger impact on in-hospital mortality in patients with CKD, while PAD gave impact in only patients with acute coronary syndrome, stable angina pectoris and old myocardial infarction except for silent ischemia.

Incidence of PAD in our study was comparable to the report from the previous reports [5,6]. However, the incidence of short-term mortality was apparently lower than the previous report [5]. The possible reasons for this discrepancy are as follows: the adverse events post PCI

**Table 3. In-hospital outcomes.**

| | All (n = 894,014) | PAD group (n = 66,891) | Non-PAD group (n = 827,123) |
|---|---|---|---|
| Primary outcome, n (%) | | | |
| In-hospital mortality | 6,206 (0.69) | 659 (0.99) | 5,547 (0.67) |
| Secondary outcomes, n (%) | | | |
| Procedure-related myocardial infarction | 4,241 (0.47) | 438 (0.65) | 3,803 (0.46) |
| Cardiac tamponade | 1,246 (0.14) | 126 (0.19) | 1,120 (0.14) |
| Cardiogenic shock | 8,039 (0.90) | 743 (1.1) | 7,296 (0.88) |
| Stent thrombosis | 1,309 (0.15) | 67 (0.10) | 1,242 (0.15) |
| Emergent surgery | 818 (0.091) | 83 (0.12) | 735 (0.089) |
| Bleeding with need for transfusion | 2,743 (0.31) | 342 (0.51) | 2,401 (0.29) |
| related to access site | 1,611 (0.18) | 204 (0.30) | 1,407 (0.17) |
| non-related to access site | 1,193 (0.13) | 147 (0.22) | 1,046 (0.13) |

PAD: Peripheral artery disease.

**Table 4. Multivariate logistic regression analysis for in-hospital mortality.**

| Variables | Odds ratio (95% CI) |
|---|---|
| PAD | 1.383 (1.251–1.528) |
| Age per 10 years | 1.455 (1.412–1.500) |
| Male | 0.759 (0.710–0.811) |
| Hypertension | 0.817 (0.765–0.873) |
| Diabetes mellitus | 1.158 (1.091–1.229) |
| Dyslipidemia | 0.629 (0.593–0.668) |
| Current smoker | 0.824 (0.768–0.885) |
| Chronic kidney disease | 1.493 (1.395–1.598) |
| Hemodialysis | 1.282 (1.149–1.431) |
| previous history of PCI | 0.841 (0.773–0.915) |
| previous history of CABG | 1.087 (0.942–1.254) |
| Chronic heart failure | 1.477 (1.364–1.598) |
| Old myocardial infarction | 1.104 (1.006–1.212) |
| Chronic obstructive pulmonary disease | 1.318 (1.127–1.540) |
| Clinical presentation at PCI | |
| STEMI | 1.000 (reference) |
| NSTEMI | 0.701 (0.641–0.766) |
| Unstable angina pectoris | 0.308 (0.279–0.341) |
| Stable angina pectoris | 0.086 (0.075–0.099) |
| Old myocardial infarction | 0.169 (0.136–0.210) |
| Silent ischemia | 0.146 (0.127–0.168) |
| Cardiac arrest within 24 hours | 3.466 (3.195–3.760) |
| Cardiogenic shock within 24 hours | 3.909 (3.579–4.268) |
| Acute heart failure within 24 hours | 1.957 (1.812–2.115) |
| Access site | |
| Femoral artery | 1.000 (reference) |
| Radial artery | 0.515 (0.481–0.552) |
| Others | 1.080 (0.963–1.212) |
| Number of diseased vessels | |
| 1 vessel | 1.000 (reference) |
| 2 vessels | 1.158 (1.075–1.249) |
| 3 vessels | 1.682 (1.555–1.819) |
| Left main disease | 2.345 (2.141–2.568) |

CABG: Coronary artery bypass grafting; NSTEMI: Non-ST elevation myocardial infarction; PAD: Peripheral artery disease; PCI: Percutaneous coronary intervention; STEMI: ST elevation myocardial infarction.

were low in Japan, which may be due to the higher rate of intravascular imaging device usage as shown in the previous reports, although the precise mechanism is still unknown [13–15]. In additions, the improvement of devices such as second-generation drug-eluting stent, guide-wire and guide-extension catheter may have contribute to the decrease of complications [16]. For example, more than half of the procedure were performed by trans-radial approach, which would contribute to the decrease of bleeding complications [17–19]. Further, in-hospital mortality has been possibly missed to be registered despite rigorous auditing system sponsored by the national professional society.

In the current study, even after the adjustment of the other variables such as coronary risk factors, PAD was independently associated with in-hospital mortality in patients who received

| | PAD (n=66891) | Non-PAD (n=827123) | Odds ratio (95%CI) | | P for interaction |
|---|---|---|---|---|---|
| **Age** | | | | | 0.73 |
| <75 | 246/34862 | 2465/519877 | 1.330 (1.124-1.574) | | |
| ≥75 | 413/32029 | 3082/307246 | 1.435 (1.268-1.624) | | |
| **Sex** | | | | | 0.090 |
| Female | 181/15340 | 1873/195514 | 1.221 (1.017-1.466) | | |
| Male | 478/51551 | 3674/631609 | 1.480 (1.313-1.669) | | |
| **Diabetes mellitus** | | | | | 0.33 |
| Yes | 373/38045 | 2570/369950 | 1.414 (1.220-1.638) | | |
| No | 286/28846 | 2977/457173 | 1.360 (1.186-1.559) | | |
| **Chronic kidney disease** | | | | | 0.005 |
| Yes | 365/25674 | 1870/132666 | 1.579 (1.370-1.821) | | |
| No | 294/41217 | 3677/694457 | 1.234 (1.076-1.416) | | |
| **Hemodialysis** | | | | | 0.89 |
| Yes | 209/14343 | 461/46402 | 1.372 (1.221-1.540) | | |
| No | 450/52548 | 5086/780721 | 1.438 (1.180-1.753) | | |
| **Chronic heart failure** | | | | | 0.50 |
| Yes | 256/15550 | 1130/104496 | 1.429 (1.261-1.619) | | |
| No | 387/51341 | 4208/722627 | 1.331 (1.129-1.569) | | |
| **Clinical presentation at PCI** | | | | | 0.002 |
| STEMI | 255/4600 | 3443/148199 | 1.249 (1.072-1.456) | | |
| NSTEMI | 114/2170 | 657/38824 | 1.522 (1.191-1.945) | | |
| Unstable angina pectoris | 106/8940 | 471/127965 | 1.361 (1.066-1.739) | | |
| Stable angina pectoris | 52/25192 | 201/308092 | 1.788 (1.292-2.473) | | |
| Old myocardial infarction | 26/3612 | 75/47124 | 2.630 (1.596-4.335) | | |
| Silent ischemia | 51/20950 | 206/141950 | 1.211 (0.8701-1.685) | | |
| **Number of diseased vessels** | | | | | 0.73 |
| 1 vessel | 180/34653 | 2100/502043 | 1.275 (1.059-1.534) | | |
| 2 vessels | 158/17894 | 1293/201039 | 1.291 (1.053-1.582) | | |
| 3 vessels | 181/10171 | 1235/91896 | 1.511 (1.250-1.828) | | |
| Left main disease | 140/4173 | 919/32145 | 1.604 (1.279-2.012) | | |
| **Access site** | | | | | 0.080 |
| Femoral artery | 433/24435 | 3813/239109 | 1.395 (1.233-1.579) | | |
| Radial artery | 136/34370 | 1376/547648 | 1.448 (1.181-1.776) | | |
| Others | 90/8086 | 358/40358 | 1.092 (0.8164-1.460) | | |

0.25  0.5  1  2  4
Odds ratio (95% CI)

**Fig 2. Impact of peripheral artery disease on in-hospital mortality in subgroups.** Interaction analysis demonstrated that PAD had the stronger impact on in-hospital mortality in patients with CKD than without CKD and PAD was not associated with an increased risk of in-hospital mortality in patients undergoing PCI for silent ischemia, whereas it consistently increased risk of in-hospital mortality in patients treated with PCI for ST elevation myocardial infarction, non-ST elevation myocardial infarction, unstable angina pectoris, stable angina pectoris and old myocardial infarction. Plots and error bars are odds ratios of PAD for in-hospital mortality and their 95% CIs. CI: Confidence interval; CKD: Chronic kidney disease; PAD: Peripheral artery disease; PCI: Percutaneous coronary intervention; STEMI: ST elevation myocardial infarction; NSTEMI: Non-ST elevation myocardial infarction.

PCI. Criqui et al. reported that PAD was an independent predictor of CAD, stroke and death even after adjustment of coronary risk factors [3]. Regarding the mechanism, although PAD is unlikely to be directly related to mortality, the presence of PAD may serve as a marker for underlying atherosclerotic processes or susceptibilities affecting other vascular beds [3]. In addition, a previous report with serial intravascular ultrasound imaging mentioned that patients with PAD demonstrated more extensive and calcified coronary atherosclerosis, impaired arterial remodeling, and greater disease progression [20]. Therefore, PAD reflects the status of progressive atherosclerosis, which would have contributed to its effectiveness as an independent predictor for in-hospital mortality in the current study. The efficacy of ABI measurement as a screening of PAD has been reported and its implementation is needed for precise risk stratification amongst PCI patients [6,21–23]. The screening of AAA and lower extremity PAD was also reported to be effective [24]. However, latest ESC guideline did no more than mention that screening for lower extremity PAD in CAD patients may be considered [25].

Only stent thrombosis was lower in the PAD group than in the Non-PAD group. It was reported that acute coronary syndrome (ACS) was a risk factor of early stent thrombosis [13,26]. In the current study, the rate of ACS was significantly lower in the PAD group than in the Non-PAD group (24.3% versus 38.5%). This difference would have contributed to the lower rate of stent thrombosis in PAD patients.

In the current study, PAD gave more impact on in-hospital mortality in patients with CKD. CKD typically represents advanced status of atherosclerosis and indeed, the incidence of major adverse event after revascularization is reported to be higher in this subgroup [27]. In terms of the indication of PCI, the presence of PAD impacted on in-hospital mortality in patients with acute coronary syndrome, stable angina pectoris and old myocardial infarction except for silent ischemia. PAD patients have less cardiac load and sometimes no symptom because of the difficulty of walking even if they complicated with CAD. Therefore, the patients who suffered adverse coronary event and had cardiac symptom even with PAD would have more progressive atherosclerosis, and they had worse in-hospital mortality than those without PAD.

## Limitations

This study has several limitations. First, PAD included lower extremity PAD and AAA in this registry and each rate was unknown. Second, this registry did not record the data of severity of lower extremity PAD and AAA. Third, this registry evaluated only in-hospital outcomes. Long-term follow-up data were unknown. Fourth, actual incidence of complications could be underestimated in this study. However, it cannot be denied that there is certain credibility to the data entry because the J-PCI registry has been linked to the board certification system in interventional cardiology. Fifth, PAD patients could be underestimated because we diagnosed them as PAD from a previous history and not all patients received ABI measurements. In particular, it was sometimes difficult to evaluate the presence of PAD for patients with cardiac arrest and cardiogenic. Sixth, the definition of bleeding complications in this study was not based on standardized definitions. Therefore, the actual incidence of bleeding complications

may be underestimated. Finally, since the J-PCI registry does not include continuous variables such as diastolic/systolic blood pressure, LDL-cholesterol level, triglyceride levels, and so on, we could not evaluate the impact of these continuous variables on the clinical outcomes.

## Conclusions

Presence of PAD was independently associated with in-hospital mortality in patients receiving PCI. However, its impact varied substantially by the patient background or indication of the procedure.

## Appendix

### Members of the Japanese association of cardiovascular intervention and therapeutics scientific committee

Kazushige Kadota (Kurashiki Central Hospital), Nobuo Shiode (Hiroshima City Hospital), Nobuhiro Tanaka (Tokyo Medical University), Tetsuya Amano (Aichi Medical University), Shiro Uemura (Kawasaki Medical School), Takashi Akasaka (Wakayama Medical University), Yoshihiro Morino (Iwate Medical University), Kenshi Fujii (Sakurabashi Watanabe Hospital), and Yutaka Hikichi (Saga University).

### Members of the Japanese association of cardiovascular intervention and therapeutics registry subcommittee

Tetsuya Amano (Aichi Medical University), Kenshi Fujii (Sakurabashi Watanabe Hospital), Shun Kohsaka (Keio University), Hideki Ishii (Nagoya University), Kengo Tanabe (Mitsui Memorial Hospital), Yukio Ozaki (Fujita Health University), Satoru Sumitsuji (Osaka University), Osamu Iida (Kansai Rosai Hospital), Hidehiko Hara (Toho University Ohashi Medical Center), Hiroaki Takashima (Aichi Medical University), Shinichi Shirai (Kokura Memorial Hospital), Mamoru Nanasato (Nagoya Daini Red Cross Hospital), Taku Inohara (Keio University), Yasunori Ueda (Osaka National Hospital), Yohei Numasawa (Japanese Red Cross Ashikaga Hospital), and Shigetaka Noma (Saiseikai Utsunomiya Hospital)

## Acknowledgments

The authors thank all the members of the CVIT and CVIT secretariat.

## Author Contributions

**Conceptualization:** Takayuki Ishihara, Osamu Iida, Shun Kohsaka, Toshiaki Mano.

**Formal analysis:** Kyohei Yamaji.

**Writing – original draft:** Takayuki Ishihara.

**Writing – review & editing:** Osamu Iida, Shun Kohsaka, Taku Inohara, Toshiro Shinke, Hirohiko Ando, Tetsuya Amano, Yasushi Sakata, Toshiaki Mano, Yuji Ikari.

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
