## [Decision Letter · Decision Letter 0]

12 Jun 2020

PONE-D-20-00744

Impact of peripheral artery disease on short-term outcomes after percutaneous coronary intervention: A report from Japanese nationwide registry

PLOS ONE

Dear Dr. Ishihara,

Thank you for submitting your manuscript to PLOS ONE. After careful consideration, we feel that it has merit but does not fully meet PLOS ONE’s publication criteria as it currently stands. Therefore, we invite you to submit a revised version of the manuscript that addresses the points raised during the review process.

We look forward to receiving your revised manuscript.

Kind regards,

Xianwu Cheng, M.D., Ph.D., FAHA

Academic Editor

PLOS ONE

Journal Requirements:

"The J-PCI registry is a registry led and supported by the Japanese Association of

Cardiovascular Intervention and Therapeutics.

KS reports investigator-initiated grant funding from Bayer and Daiichi Sankyo, and

personal fees from Bayer, AstraZeneca and Bristol-Myers Squibb. TI has a research

grant from Boston Scientific. TS receives Remuneration from Abbott Vascular, Dai-ichi

Sankyo, Bayer, Sanofi, and Nipro. TA receives lecture fees from Astellas Pharma,

AstraZeneca, Bayer, Daiichi Sankyo, and Bristol-Myers Squibb. YS receives

Honorarium from·Otsuka Pharmaceutical, Daiichi Sankyo, Takeda

Pharmaceutical,·Mitsubishi Tanabe Pharma, Medtronic Japan and Boehringer

Ingelheim Japan, research grant from Edwards Lifesciences, FUJIFILM RI

Pharma,·REGiMMUNE, and·Roche Diagnostics, and Scholarship (educational)

grant/endowed chair from Otsuka Pharmaceutical, Johnson & Johnson, St. Jude

Medical Japan, Daiichi Sankyo,·Takeda Pharmaceutical,·Mitsubishi Tanabe Pharma,

Teijin Pharma Limited,·Boehringer Ingelheim Japan, Bayer Yakuhin, BIOTRONIK

Japan, Boston Scientific, Medtronic Japan. TM has a research grant from Abbott

vascular Japan. YI has research grants from Boston Scientific, Asahi intech, Nipro,

Sanofi, Daiichi-Sankyo, and Terumo, and receives lecture fee from Asteras Amgen

Biopharma, Astrazeneca, Abbott vascular, Sanofi, Daiichi-Sankyo, Boehringer-

Ingerheim, Bayer, and Bristol Meyers-Squib. The remaining authors have no

disclosures to report.

The funders had no role in study design, data collection and analysis, decision to

publish, or preparation of the manuscript."

We note that you received funding from a commercial source: "Bayer, Daiichi Sankyo, AstraZeneca, Abbott Vascular, Sanofi, Nipro, Astellas Pharma, Otsuka Pharmaceutica, Takeda Pharmaceutical,·Mitsubishi Tanabe Pharma, Medtronic Japan, Boehringer Ingelheim, Edwards Lifesciences, FUJIFILM RI Pharma,·REGiMMUNE, ·Roche Diagnostics, Johnson & Johnson, Bayer Yakuhin, Teijin Pharma Limited,BIOTRONIK, Medtronic, Asahi intech, Asteras Amgen Biopharma, Astrazeneca, Abbott vascular, Terumo and Bristol-Myers Squibb

Additional Editor Comments (if provided):

Although this study is huge scale retrospective study, as As you will gather from the reviews, both reviewers have concerned the methodological proplems, data collection, presentation, and statistical analysis. One pointed out that conclusions were not fully supported by their findings. The academic editor concurs. We also found that one of the major weakess is study design. Therefore, the authors may resubmit a revised version but it will be re-reviewed and there exists no guarantee that even with revision it will necessarily be accepted.

Reviewers' comments:

Reviewer's Responses to Questions

**Comments to the Author**

1. Is the manuscript technically sound, and do the data support the conclusions?

Reviewer #1: Yes

Reviewer #2: Partly

2. Has the statistical analysis been performed appropriately and rigorously? 

Reviewer #1: No

Reviewer #2: Yes

3. Have the authors made all data underlying the findings in their manuscript fully available?

Reviewer #1: No

Reviewer #2: Yes

4. Is the manuscript presented in an intelligible fashion and written in standard English?

Reviewer #1: Yes

Reviewer #2: Yes

5. Review Comments to the Author

Reviewer #1: The authors investigated the impact of peripheral artery disease on in-hospital mortality using a large nationwide registry. Results from their analysis (table 4) show that peripheral artery disease is independently associated with in-hospital mortality.

The authors have access to a huge dataset n=894,014 with many variables and outcomes. Since the number of observations is huge, it is possible to fit complex models to the data. One can, for instance, explore non-linear effects or many interactions between risk-factors. Based on table 4, however, a simple additive model with linear effects was fitted to the data. In addition, continuous risk-factors (e.g. blood-pressure) were dichotomized. I believe it is better to leave the continuous variables untouched as dichotomization leads to all kinds of problems (e.g. assumption of sudden effect, information loss, use of arbitrary cut-offs) https://onlinelibrary.wiley.com/doi/abs/10.1002/0470011815.b2a10012.

I suggest the authors do not categorize the risk-factors and explore more complex models with non-linear effects.

Page 2 line 11: What does the number after the +- symbol mean? Please explain in the manuscript.

Page 2 line 11: Are these the results from the multi-variable model? Please mention in the manuscript.

Pages 27 and 28 tables 3 and 4: Are some of the hospital outcomes not predictive for mortality? If a risk-factor as 'Acute heart failure within 24 hours' is added to the model, why not 'Cardiogenic shock'?

All tables: Why report p-values with this number of observations. P-values are not really relevant.

Page 28 table 4: Please change the unit of age (e.g.\\ per 10 years). The effect OR: 1.04 (95\\%CI: 1.04-1.04) is not really informative.

Reviewer #2: The article is certainly well written and attractive. The authors investigated the patients who undergoing PCI with or without PAD have some different clinical events and PAD was mainly affect in-hospital mortality. The findings of this paper is very interestiong. Nevertheless, there still have a number of issues that need to be clarified.

1. The authors demonstrated that “This is the first large-scale report......undergoing PCI”, I also agree with authors findings, even though, the authors do not fully explained how and what reason could induced such condition.

2. Page 9, line 17, the authors mentioned that “The incidences of procedure-related myocardial infarction..... were also significantly higher in PAD group”, But, according to your data in table 3, those variables were higher in non-PAD group, is this just a typing error or misunderstanding? As common knowledge, such as “procedure related myocardial infarction” is more related with coronary artery condition, like as severity of atherosclerosis, calcification, lesion length and plaque burden, even related with the operator’s skill and experience, I really do not understand why the procedure related complication correlated with PAD or not.

3. Page 9, sentence 18, “However, only stent thrombosis was significantly lower in PAD group”, normally, stent thrombosis was related with stent length, under expension and malapposion of stent and so on. Would you explain why the other complication such as cardiogenic shock was higher in PAD group and stent thrombosis was lower in PAD group?

4. Page 12, sentence 1, “more than half of the procedure was performed by trans-radial approach..... decrease of bleeding complications”. But why there were such high rate of access site related bleeding in two groups?

5. Page 14, line 4, “ …affected…” should replace with affect.

6. Table 2, line 12, “Bioresorbable scattfold” is inappropriate word and have spelling error, should replace to bioabsorbable scaffold.

6. PLOS authors have the option to publish the peer review history of their article (what does this mean?). If published, this will include your full peer review and any attached files.

Reviewer #1: No

Reviewer #2: No

---

## [Author Response · Author response to Decision Letter 0]

17 Jul 2020

Responses to the Editor

>> We appreciate the time and expertise of the editors and reviewers and have revised our manuscript according to their suggestions.

Journal Requirements:

>> Thank you for your comment. We adjusted the manuscript to meet the style requirements of PLOS ONE.

2. Thank you for stating the following in the Financial Disclosure section: "The J-PCI registry is a registry led and supported by the Japanese Association of Cardiovascular Intervention and Therapeutics. KS reports investigator-initiated grant funding from Bayer and Daiichi Sankyo, and personal fees from Bayer, AstraZeneca and Bristol-Myers Squibb. TI has a research grant from Boston Scientific. TS receives Remuneration from Abbott Vascular, Dai-ichi Sankyo, Bayer, Sanofi, and Nipro. TA receives lecture fees from Astellas Pharma, AstraZeneca, Bayer, Daiichi Sankyo, and Bristol-Myers Squibb. YS receives Honorarium from·Otsuka Pharmaceutical, Daiichi Sankyo, Takeda Pharmaceutical,·Mitsubishi Tanabe Pharma, Medtronic Japan and Boehringer Ingelheim Japan, research grant from Edwards Lifesciences, FUJIFILM RI Pharma,·REGiMMUNE, and·Roche Diagnostics, and Scholarship (educational) grant/endowed chair from Otsuka Pharmaceutical, Johnson & Johnson, St. Jude Medical Japan, Daiichi Sankyo,·Takeda Pharmaceutical,·Mitsubishi Tanabe Pharma, Teijin Pharma Limited,·Boehringer Ingelheim Japan, Bayer Yakuhin, BIOTRONIK Japan, Boston Scientific, Medtronic Japan. TM has a research grant from Abbott vascular Japan. YI has research grants from Boston Scientific, Asahi intech, Nipro, Sanofi, Daiichi-Sankyo, and Terumo, and receives lecture fee from Asteras Amgen Biopharma, Astrazeneca, Abbott vascular, Sanofi, Daiichi-Sankyo, Boehringer-Ingerheim, Bayer, and Bristol Meyers-Squib. The remaining authors have no disclosures to report. The funders had no role in study design, data collection and analysis, decision to publish, or preparation of the manuscript."

We note that you received funding from a commercial source: "Bayer, Daiichi Sankyo, AstraZeneca, Abbott Vascular, Sanofi, Nipro, Astellas Pharma, Otsuka Pharmaceutica, Takeda Pharmaceutical,·Mitsubishi Tanabe Pharma, Medtronic Japan, Boehringer Ingelheim, Edwards Lifesciences, FUJIFILM RI Pharma,·REGiMMUNE, ·Roche Diagnostics, Johnson & Johnson, Bayer Yakuhin, Teijin Pharma Limited,BIOTRONIK, Medtronic, Asahi intech, Asteras Amgen Biopharma, Astrazeneca, Abbott vascular, Terumo and Bristol-Myers Squibb”. 

Please provide an amended Competing Interests Statement that explicitly states this commercial funder, along with any other relevant declarations relating to employment, consultancy, patents, products in development, marketed products, etc. Within this Competing Interests Statement, please confirm that this does not alter your adherence to all PLOS ONE policies on sharing data and materials by including the following statement: "This does not alter our adherence to PLOS ONE policies on sharing data and materials.” (as detailed online in our guide for authors http://journals.plos.org/plosone/s/competing-interests). If there are restrictions on sharing of data and/or materials, please state these. Please note that we cannot proceed with consideration of your article until this information has been declared. Please include your amended Competing Interests Statement within your cover letter. We will change the online submission form on your behalf.

>> Thank you for your suggestion. We included our amended Competing Interests Statement within our cover letter. 

Additional Editor Comments (if provided):

Although this study is huge scale retrospective study, as you will gather from the reviews, both reviewers have concerned the methodological problems, data collection, presentation, and statistical analysis.

>> Reviewer #1 raised a concern regarding the treatment of continuous variables in our statistical models (dichotomization of continuous variables). However, in our study, variables such as age were not dichotomized but rather were treated as continuous variables throughout the process. We have re-emphasized this point in the revised manuscript. In addition, the reviewer suggested that “cardiogenic shock” should be introduced into the multivariate model, and this variable was incorporated into our new model. Also, in regard to the tables, the reviewer suggested that the P-values be deleted and the unit of age be changed to 10 years. Both these suggestions have been fully implemented.

One pointed out that conclusions were not fully supported by their findings. The academic editor concurs. 

>> We see that Reviewer #2 commented that the “authors did not fully explain how and what reason could induced such condition (PAD independently associated with in-hospital mortality in PCI patients)”. We agree that PAD is not directly related to mortality, although the presence of PAD is known to act as a surrogate for underlying atherosclerotic processes or susceptibilities affecting other vascular beds (Circ Res 2015;116:1509-1526). A previous report with serial intravascular ultrasound imaging mentioned that patients with PAD demonstrated more extensive and calcified coronary atherosclerosis, impaired arterial remodeling, and greater disease progression (J Am Coll Cardiol. 2011;57:1220-5). We concluded that PAD may reflect the status of progressive atherosclerosis, which would contribute to its effectiveness as an independent predictor of in-hospital mortality in the current study. We modified the Discussion section to include these points.

Reviewer #2 also expressed concern regarding the incidence of procedure-related myocardial infarction. Specifically, the reviewer wondered whether the incidence was higher in Non-PAD patients). However, we can confidently state that this was not the case. The adverse outcomes were consistently higher in the PAD group than in the Non-PAD group. 

- In-hospital mortality : 0.99% vs. 0.67%

- Procedure-related myocardial infarction : 0.65% vs. 0.46%

- Cardiac tamponade : 0.19% vs. 0.14%

- Cardiogenic shock : 1.1% vs. 0.88%

- Emergent surgery : 0.12% vs. 0.089%

- Bleeding with need for transfusion : 0.51% vs. 0.29%

Reviewer #2 also expressed concern regarding the lower incidence of stent thrombosis in the PAD group. It was reported that acute coronary syndrome (ACS) was a risk factor of early stent thrombosis (Circulation 2012;125:584-591, Eur Heart J 2018;39:213-260). In the current study, the rate of ACS was significantly lower in the PAD group than in the Non-PAD group (24.3% versus 38.5%). This difference would have contributed to the lower rate of stent thrombosis in PAD patients.

Lastly, the reviewers were concerned that access site-related bleeding was high in both the PAD and Non-PAD groups. However, in the current study the incidence of access site-related bleeding was 0.30% in the PAD group and 0.17% in the Non-PAD group, and these values were not much higher than those in the previous studies (Radial: 0.10-2.6%, Femoral: 0.20-6.8% [J Am Coll Cardiol 2012;60:2481-9, J Am Coll Cardiol 2012;60:2490-9]).

We believe that our revisions have addressed the concerns and issues raised by the reviewers. However, please do let us know if there are any additional concerns.

 

Responses to Reviewer #1

#1: The authors have access to a huge dataset n=894,014 with many variables and outcomes. Since the number of observations is huge, it is possible to fit complex models to the data. One can, for instance, explore non-linear effects or many interactions between risk-factors. Based on table 4, however, a simple additive model with linear effects was fitted to the data. In addition, continuous risk-factors (e.g. blood-pressure) were dichotomized. I believe it is better to leave the continuous variables untouched as dichotomization leads to all kinds of problems (e.g. assumption of sudden effect, information loss, use of arbitrary cut-offs) https://onlinelibrary.wiley.com/doi/abs/10.1002/0470011815.b2a10012.

I suggest the authors do not categorize the risk-factors and explore more complex models with non-linear effects.

>> Thank you for your important suggestions. Please note that the continuous variable (age) was indeed treated as a continuous variable in our model. As for your suggestion that we explore non-linear relationships amongst the variables, this was not possible because the remaining variables were all categorical variables (unfortunately, the J-PCI does not include blood pressure or other numerical variables). We added the following sentences to the Methods and Discussion sections.

(Added sentence)

Methods section (page R8, lines 9-10): The J-PCI Registry does not include blood pressure or the other numerical variables except for age.

Discussion section (page R18, lines 11-13): Seventh, since all the variables except for age were categorical variables, it was not possible to explore non-linear relationships amongst the variables.

#2: Page 2 line 11: What does the number after the +- symbol mean? Please explain in the manuscript.

>> Thank you for your question. It is a very important point and I am sorry that we did not properly explain it. The ± symbol indicated the standard deviation. We revised the relevant sentence in the Statistical analysis section.

(Before revision)

All results are expressed as mean ± SD unless otherwise stated.

(After revision)

Methods section (page R8, line 10): All results are expressed as the mean ± standard deviation unless otherwise stated.

#3: Page 2 line 11: Are these the results from the multi-variable model? Please mention in the manuscript.

>> Thank you for your question. As mentioned in the Methods section of the Abstract, the primary outcome was in-hospital mortality, and multivariable modeling was performed. The crude incidences were compared between the PAD and the Non-PAD groups, and crude in-hospital mortality rate was higher in the PAD group than in the Non-PAD group (0.99% vs. 0.67%). Then, multivariable analysis for in-hospital mortality was performed by logistic regression analysis.

#4: Pages 27 and 28 tables 3 and 4: Are some of the hospital outcomes not predictive for mortality? If a risk-factor as 'Acute heart failure within 24 hours' is added to the model, why not 'Cardiogenic shock'?

>> Thank you for your important question. As you pointed out, cardiac arrest within 24 hours and acute heart failure within 24 hours are independent predictors for in-hospital mortality. As suggested, “Cardiogenic shock” was included in our new model. We revised the contents related to the new model as follows: 

(Before revision)

PAD was associated with an increased risk of in-hospital mortality (odds ratio [OR] 1.40 [95% confidence interval 1.26-1.55]). However, the impact of PAD differed by kidney condition (OR 1.59 [1.38-1.83] for patients with chronic kidney disease [CKD] and OR 1.26 [1.10-1.45] without CKD: P for interaction 0.005), and by clinical presentation: PAD was not associated with an increased risk of in-hospital mortality in patients undergoing PCI for silent ischemia (OR 1.18 [0.85-1.64]: P for interaction 0.002).

(After revision)

Abstract session (page R2, lines 15 – page R3, line 3): PAD was associated with an increased risk of in-hospital mortality (odds ratio [OR] 1.383 [95% confidence interval 1.251-1.528]). However, the impact of PAD differed by kidney condition (OR 1.578 [1.370-1.821] for patients with chronic kidney disease [CKD] and OR 1.234 [1.076-1.416] without CKD: P for interaction 0.005), and by clinical presentation: PAD was not associated with an increased risk of in-hospital mortality in patients undergoing PCI for silent ischemia (OR 1.211 [0.8701-1.685]: P for interaction 0.002).

(Before revision)

Multivariable analysis for in-hospital mortality was performed by logistic regression analysis and the following explanatory variables were included: PAD, age, sex, hypertension, diabetes mellitus, dyslipidemia, current smoker, CKD, hemodialysis, previous history of PCI, previous history of coronary artery bypass grafting (CABG), chronic heart failure, previous myocardial infarction, chronic obstructive pulmonary disease, clinical presentation at the time of PCI, cardiac arrest within 24 hours, acute heart failure within 24 hours, access site, number of diseased vessels.

(After revision)

Methods section (page R8, line 12 – page R9, line 1): Multivariable analysis for in-hospital mortality was performed by logistic regression analysis and the following explanatory variables were included: PAD, age, sex, hypertension, diabetes mellitus, dyslipidemia, current smoker, CKD, hemodialysis, previous history of PCI, previous history of coronary artery bypass grafting (CABG), chronic heart failure, previous myocardial infarction, chronic obstructive pulmonary disease, clinical presentation at the time of PCI, cardiac arrest within 24 hours, cardiogenic shock within 24 hours, acute heart failure within 24 hours, access site, and number of diseased vessels.

(Before revision)

After the adjustment for confounders, PAD was associated with an increased risk of in-hospital mortality (Table 4: odds ratio [OR] 1.40 [95% confidence interval 1.26-1.55], P<0.001). Similar to PAD, chronic kidney disease (OR 1.58 [1.47-1.69]), cardiac arrest within 24 hours (OR 7.34 [6.80-7.94]), acute heart failure within 24 hours (OR 4.13 [3.85-4.44]), and left main disease (OR 2.75 [2.50-3.02]) demonstrated relatively higher odds ratio.

(After revision)

Results section (page R12, line 8 - page R13 line 2): After the adjustment for confounders, PAD was associated with an increased risk of in-hospital mortality (Table 4: odds ratio [OR] 1.383 [95% confidence interval 1.251-1.528]). Similar to PAD, chronic kidney disease (OR 1.493 [1.395-1.598]), cardiac arrest within 24 hours (OR 3.466 [3.195-3.760]), cardiogenic shock within 24 hours (OR 3.909 [3.579-4.268]), acute heart failure within 24 hours (OR 1.957 [1.812-2.115]), and left main disease (OR 2.345 [2.141-2.568]) demonstrated relatively higher odds ratios.

(Revised Table)

page R13-14, Table 4

(Before revision)

The magnitude of impact of PAD on in-hospital mortality was different by presence or absence of CKD (Fig 2: OR 1.59 [1.38-1.83] and OR 1.26 [1.10-1.45] for CKD and no CKD subgroup, respectively: P for interaction=0.005). In addition, PAD was not associated with an increased risk of in-hospital mortality in patients undergoing PCI for silent ischemia (OR 1.18 [0.85-1.64]), whereas it consistently increased risk of in-hospital mortality in patients treated with PCI for ST elevation myocardial infarction (OR 1.26 [1.09-1.47], non-ST elevation myocardial infarction (OR 1.60 [1.26-2.04]), unstable angina pectoris (OR 1.40 [1.10-1.79]) , stable angina pectoris (OR 1.78 [1.29-2.46]) and old myocardial infarction (OR 2.68 [1.58-4.29], P for interaction=0.002).

(After revision)

Results section (page R14, lines 4-13): The magnitude of impact of PAD on in-hospital mortality differed according to the presence or absence of CKD (Fig. 2: OR 1.579 [1.370-1.821] and OR 1.234 [1.076-1.416] for the CKD and the No-CKD subgroup, respectively: P for interaction=0.005). In addition, PAD was not associated with an increased risk of in-hospital mortality in patients undergoing PCI for silent ischemia (OR 1.211 [0.8701-1.685]), whereas it consistently increased the risk of in-hospital mortality in patients treated with PCI for ST elevation myocardial infarction (OR 1.249 [1.072-1.456], non-ST elevation myocardial infarction (OR 1.522 [1.191-1.945]), unstable angina pectoris (OR 1.361 [1.066-1.739]), stable angina pectoris (OR 1.788 [1.292-2.473]) and old myocardial infarction (OR 2.630 [1.596-4.335], P for interaction=0.002).

#5: All tables: Why report p-values with this number of observations. P-values are not really relevant.

>> Thank you for your important suggestion. As suggested, we deleted the P-values from the outcomes.

(Before revision)

A two-sided p-value of <0.05 was considered significant for all tests.

(After revision)

Methods section (page R9, lines 3-4): A two-sided p-value of <0.05 was considered significant in the interaction analysis.

(Revised Table)

page R9-10, Table 1

page R11, Table 2

page R12, Table 3

page R13-14, Table 4

(Revised Figure)

Figure 2

#6: Page 28 table 4: Please change the unit of age (e.g.\\ per 10 years). The effect OR: 1.04 (95\\%CI: 1.04-1.04) is not really informative.

>> Thank you for your important suggestion. As suggested, the original units were not very informative. We revised the unit of age to per 10 years as recommended.

(Revised Table)

page R13-14, Table 4 

Responses to Reviewer #2

1. The authors demonstrated that “This is the first large-scale report......undergoing PCI”, I also agree with authors findings, even though, the authors do not fully explained how and what reason could induced such condition.

>> Thank you for your important comment. We agree that PAD is probably not directly related to mortality, and the presence of PAD perhaps acts as a surrogate for underlying atherosclerotic processes or susceptibilities affecting other vascular beds (Circ Res 2015;116:1509-1526). A previous report with serial intravascular ultrasound imaging mentioned that patients with PAD demonstrated more extensive and calcified coronary atherosclerosis, impaired arterial remodeling, and greater disease progression (J Am Coll Cardiol. 2011;57:1220-5). We concluded that PAD may reflect the status of progressive atherosclerosis, and thus may have acted as an independent predictor of in-hospital mortality in the current study. The Discussion section of the manuscript has been modified accordingly.

(Before revision)

Although PAD is unlikely to be directly related to mortality, the presence of PAD may serve as a marker for underlying atherosclerotic processes or susceptibilities affecting other vascular beds.

(After revision)

Discussion section (page R16, lines 6-13): Regarding the mechanism, although PAD is unlikely to be directly related to mortality, the presence of PAD may serve as a marker for underlying atherosclerotic processes or susceptibilities affecting other vascular beds [3]. In addition, a previous report with serial intravascular ultrasound imaging mentioned that patients with PAD demonstrated more extensive and calcified coronary atherosclerosis, impaired arterial remodeling, and greater disease progression [20]. Therefore, PAD reflects the status of progressive atherosclerosis, which would have contributed to its effectiveness as an independent predictor for in-hospital mortality in the current study.

(Added reference)

20. Hussein AA, Uno K, Wolski K, Kapadia S, Schoenhagen P, Tuzcu EM, et al. Peripheral arterial disease and progression of coronary atherosclerosis. J Am Coll Cardiol. 2011;57:1220-5.

2. Page 9, line 17, the authors mentioned that “The incidences of procedure-related myocardial infarction..... were also significantly higher in PAD group”, But, according to your data in table 3, those variables were higher in non-PAD group, is this just a typing error or misunderstanding? As common knowledge, such as “procedure related myocardial infarction” is more related with coronary artery condition, like as severity of atherosclerosis, calcification, lesion length and plaque burden, even related with the operator’s skill and experience, I really do not understand why the procedure related complication correlated with PAD or not.

>> Thank you for your comment. However, Table 3 illustrates that the listed outcome variables were indeed higher in the PAD group than in the Non-PAD group:

- In-hospital mortality : 0.99% vs. 0.67%

- Procedure-related myocardial infarction : 0.65% vs. 0.46%

- Cardiac tamponade : 0.19% vs. 0.14%

- Cardiogenic shock : 1.1% vs. 0.88%

- Emergent surgery : 0.12% vs. 0.089%

- Bleeding with need for transfusion : 0.51% vs. 0.29%

We concur that coronary artery conditions such as severity of atherosclerosis, calcification, lesion length and plaque burden as well as the operator’s skill and experience would be related to the procedure-related myocardial infarction. PAD is unlikely to be directly related to procedure-related myocardial infarction, but we believe that the presence of PAD would reflect the progressive status of other vascular beds, including coronary artery disease (J Am Coll Cardiol. 2011;57:1220-5, Circ Res 2015;116:1509-1526.). 

3. Page 9, sentence 18, “However, only stent thrombosis was significantly lower in PAD group”, normally, stent thrombosis was related with stent length, under expension and malapposion of stent and so on. Would you explain why the other complication such as cardiogenic shock was higher in PAD group and stent thrombosis was lower in PAD group?

>> Thank you for you important question. As you pointed out, only stent thrombosis was lower in the PAD group than in the Non-PAD group. Acute coronary syndrome (ACS) has been reported as a risk factor of early stent thrombosis (Circulation 2012;125:584-591, Eur Heart J 2018;39:213-260). In the current study, the rate of ACS was significantly lower in the PAD group than in the Non-PAD group (24.3% versus 38.5%). This difference would have contributed to the lower rate of stent thrombosis in PAD patients. We added this point to the Discussion section as follows:

(Added sentences)

Discussion section (page R17, lines 1-5): Only stent thrombosis was lower in the PAD group than in the Non-PAD group. It was reported that acute coronary syndrome (ACS) was a risk factor of early stent thrombosis [13, 26]. In the current study, the rate of ACS was significantly lower in the PAD group than in the Non-PAD group (24.3% versus 38.5%). This difference would have contributed to the lower rate of stent thrombosis in PAD patients.

(Added reference)

26. Valgimigli M, Bueno H, Byrne RA, Collet JP, Costa F, Jeppsson A, et al. 2017 ESC focused update on dual antiplatelet therapy in coronary artery disease developed in collaboration with EACTS: The Task Force for dual antiplatelet therapy in coronary artery disease of the European Society of Cardiology (ESC) and of the European Association for Cardio-Thoracic Surgery (EACTS). Eur Heart J 2018; 39: 213–260.

4. Page 12, sentence 1, “more than half of the procedure was performed by trans-radial approach..... decrease of bleeding complications”. But why there were such high rate of access site related bleeding in two groups?

>> Thank you for your important comment. However, the incidence of access site-related bleeding was 0.30% in the PAD group and 0.17% in the Non-PAD group in the current study, and these values were not much higher than those from the previous studies (Radial: 0.10-2.6%, Femoral: 0.20-6.8% [ J Am Coll Cardiol 2012;60:2481-9, J Am Coll Cardiol 2012;60:2490-9]).

5. Page 14, line 4, “ …affected…” should replace with affect.

>> Thank you catching this careless mistake. We corrected it.

(Before revision)

Finally, even with the multivariate analysis, unmeasured and residual confounding could affected the results.

(After revision)

Discussion section (page R18, lines 13-14): Finally, even with the multivariate analysis, unmeasured and residual confounding could affect the results.

6. Table 2, line 12, “Bioresorbable scattfold” is inappropriate word and have spelling error, should replace to bioabsorbable scaffold.

>> Thank you for your suggestion. We modified this phrase as follows:

(Before revision) Bioresorbable scattfold, n (%)

(After revision)

page R 11, Table 2: Bioabsorbable scaffold, n (%)

---

## [Decision Letter · Decision Letter 1]

18 Aug 2020

PONE-D-20-00744R1

Impact of peripheral artery disease on short-term outcomes after percutaneous coronary intervention: A report from Japanese nationwide registry

PLOS ONE

Dear Dr. Ishihara

Thank you for submitting your manuscript to PLOS ONE. After careful consideration, we feel that it has merit but does not fully meet PLOS ONE’s publication criteria as it currently stands. Therefore, we invite you to submit a revised version of the manuscript that addresses the points raised during the review process.

Please submit your revised manuscript by September 20, 2020 If you will need more time than this to complete your revisions, please reply to this message or contact the journal office at plosone@plos.org. Please include the following items when submitting your revised manuscript:

We look forward to receiving your revised manuscript.

Kind regards,

Xianwu Cheng, M.D., Ph.D., FAHA

Academic Editor

PLOS ONE

Additional Editor Comments (if provided):

The Reviewer#1 has still concerned several important issues.

Reviewers' comments:

Reviewer's Responses to Questions

**Comments to the Author**

1. If the authors have adequately addressed your comments raised in a previous round of review and you feel that this manuscript is now acceptable for publication, you may indicate that here to bypass the “Comments to the Author” section, enter your conflict of interest statement in the “Confidential to Editor” section, and submit your "Accept" recommendation.

Reviewer #1: (No Response)

Reviewer #2: All comments have been addressed

2. Is the manuscript technically sound, and do the data support the conclusions?

Reviewer #1: Partly

Reviewer #2: Partly

3. Has the statistical analysis been performed appropriately and rigorously? 

Reviewer #1: No

Reviewer #2: N/A

4. Have the authors made all data underlying the findings in their manuscript fully available?

Reviewer #1: Yes

Reviewer #2: Yes

5. Is the manuscript presented in an intelligible fashion and written in standard English?

Reviewer #1: Yes

Reviewer #2: Yes

6. Review Comments to the Author

Reviewer #1: (No Response)

Reviewer #2: (No Response)

7. PLOS authors have the option to publish the peer review history of their article (what does this mean?). If published, this will include your full peer review and any attached files.

Reviewer #1: No

Reviewer #2: No

---

## [Author Response · Author response to Decision Letter 1]

21 Aug 2020

Responses to the Editor

>> We appreciate the time and expertise of the editors and reviewers and have revised our manuscript according to their suggestions.

1. Reviewer 1 raised a concern regarding the treatment of continuous variables. I think this is still an important, unsolved point. On page 7 of the manuscript, the authors mention a large number of continuous variables (e.g. diastolic/systolic blood pressure, LDL-cholesterol level, triglyceride levels,. . .) that are all dichotomized into clinical fenotypes such as hypertension or dyslipidemia. If possible, I suggest that the authors use the continuous variables instead of the clinical fenotypes. I might be possible, for instance, that only LDL-cholesterol levels impact the mortality risk and not the triglyceride levels. This distinction is lost if you use the clincial fenotypes. This is a missed opportunity, especially since the dataset contains a large number of observations.

If these continuous variables are not available, please mention this in the discussion section.

>> Thank you for your important suggestion. Unfortunately, the J-PCI registry does not include continuous variables such as diastolic/systolic blood pressure, LDL-cholesterol level, triglyceride levels, and so on (these variables were used by the local investigators/ data managers to define hypertension and dyslipidemia, but the numbers themselves were not recorded). As suggested by the editor, we mentioned this in the discussion section.

(Added sentence)

Discussion section (page R18, lines 10-13): Finally, since the J-PCI registry does not include continuous variables such as diastolic/systolic blood pressure, LDL-cholesterol level, triglyceride levels, and so on, we could not evaluate the impact of these continuous variables on the clinical outcomes.

2. The +- symbol is used in the abstract without explanation. Please remove this or mention what it means.

>> Thank you for pointing it out. The ± symbol indicated the standard deviation. We revised the sentence as follows:

(Before revision)

For the present study, we extracted data of 894,014 percutaneous coronary intervention (PCI) cases performed between 2014 and 2017 (mean age 70.2±11.0 years).

(After revision)

Abstract section (page R2, lines 9-11): For the present study, we extracted data of 894,014 percutaneous coronary intervention (PCI) cases performed between 2014 and 2017 (mean age 70.2 years [standard deviation 11.0]).

3. Table 1: Remove the column with 'all patients'.

>> Thank you for pointing it out. As suggested, we removed the column with `all patients`. 

(Revised Table)

Results section (page R9-10): Table 1

3. Table 4: Round all odds ratios up to 2 or 3 decimals.

>> Thank you for pointing it out. As suggested, we rounded off all odds rations to three decimal places.

(Revised Table)

Results section (page R13-14): Table 4

4. The added sentences at the end of the limitations on page 18 should be removed. Even with only 1 continuous variable, it is possible to explore non-linear associations. In addition, what do the authors want to say with Finally, even with the multivariate analysis, unmeasured and residual confounding could a ect the results? I would remove this sentence.

>> Thank you for important suggestions. As suggested, we removed these sentences.

(Removed sentences)

Discussion section: Seventh, since all the variables except for age were categorical variables, it was not possible to explore non-linear relationships amongst the variables. Finally, even with the multivariate analysis, unmeasured and residual confounding could affect the results.

---

## [Decision Letter · Decision Letter 2]

21 Sep 2020

Impact of peripheral artery disease on short-term outcomes after percutaneous coronary intervention: A report from Japanese nationwide registry

PONE-D-20-00744R2

Dear Dr Ishihara

We’re pleased to inform you that your manuscript has been judged scientifically suitable for publication and will be formally accepted for publication once it meets all outstanding technical requirements.

Kind regards,

Xianwu Cheng, M.D., Ph.D., FAHA

Academic Editor

PLOS ONE

Additional Editor Comments (optional):

None.

Reviewers' comments:

Reviewer's Responses to Questions

**Comments to the Author**

1. If the authors have adequately addressed your comments raised in a previous round of review and you feel that this manuscript is now acceptable for publication, you may indicate that here to bypass the “Comments to the Author” section, enter your conflict of interest statement in the “Confidential to Editor” section, and submit your "Accept" recommendation.

Reviewer #1: All comments have been addressed

2. Is the manuscript technically sound, and do the data support the conclusions?

Reviewer #1: Yes

3. Has the statistical analysis been performed appropriately and rigorously? 

Reviewer #1: Yes

4. Have the authors made all data underlying the findings in their manuscript fully available?

Reviewer #1: Yes

5. Is the manuscript presented in an intelligible fashion and written in standard English?

Reviewer #1: Yes

6. Review Comments to the Author

Reviewer #1: All comments have been answered. Although I believe the paper can be accepted, I still have some doubts whether the results are new or interesting. Unfortunately, I am not familiar with the field of heart disease so I cannot evaluate this.

7. PLOS authors have the option to publish the peer review history of their article (what does this mean?). If published, this will include your full peer review and any attached files.

Reviewer #1: No

---

## [Editor Report · Acceptance letter]

24 Sep 2020

PONE-D-20-00744R2 

Impact of peripheral artery disease on short-term outcomes after percutaneous coronary intervention: A report from Japanese nationwide registry 

Dear Dr. Ishihara:

I'm pleased to inform you that your manuscript has been deemed suitable for publication in PLOS ONE. Congratulations! Your manuscript is now with our production department. 

Kind regards, 

on behalf of

Associate Prof. Xianwu Cheng 

Academic Editor

PLOS ONE